# GEOMETRIC INDUCTIVE BIASES FOR DIFFUSION-BASED GRAPH GENERATION

**Florian Grötschla, Saku Peltonen, Anisha Mohamed Sahabdeen, Roger Wattenhofer**
ETH Zurich, Switzerland
{fgroetschla,speltonen}@ethz.ch

## ABSTRACT

We view graph generation as the problem of sampling a geometric configuration of nodes in a latent space whose geometry represents the structure of the underlying graph. In this framework, a graph is generated by first sampling node embeddings through a diffusion process defined on the latent geometry, and then recovering edges from the resulting configuration based on geometric relations. The choice of latent geometry acts as the primary inductive bias of the model and determines which graph structures can be represented naturally. By instantiating this process with latent spaces of different curvature, we analyze how geometric assumptions influence generation quality. Experiments on synthetic graph families and molecular graphs show that performance depends systematically on curvature and is highest when the latent geometry matches the structural properties of the data. These results indicate that graph generation can be effectively guided by latent geometry alone, without relying on node features or domain-specific constraints.

## 1 INTRODUCTION

Graph generation aims to model distributions over discrete relational structures and to generate new graphs that reflect the statistical characteristics of observed data. Applications span molecular modeling, biological and social networks, and program analysis, where graphs exhibit irregular connectivity, variable size, and heterogeneous topology (Faez et al., 2021; Teji et al., 2024). Unlike grid-structured data, graphs are permutation-invariant and lack a natural ordering or regular coordinate system, posing unique challenges for generative modeling. Existing graph generative models include autoregressive approaches such as GraphRNN (You et al., 2018) and GRAN (Liao et al., 2020), flow- and VAE-based molecular generators (Jin et al., 2018; Madhawa et al., 2019), and diffusion-based models (Vignac et al., 2023). While these methods achieve strong empirical performance, they typically operate directly on discrete graph representations or rely on Euclidean latent spaces, making the inductive biases governing generation largely implicit and difficult to analyze. In this work, we adopt a geometric formulation in which graphs are represented as configurations of node embeddings in a latent metric space. Structural relations are encoded through distances, and edges are recovered via a simple geometric decoding rule (Figure 1). The choice of latent geometry becomes an explicit inductive bias, determining which structural patterns can be represented naturally. While non-Euclidean latent spaces have been studied extensively for discriminative graph tasks such as node classification and link prediction (Ganea et al., 2018; Chami et al., 2019; Mathieu & Nickel, 2020; Steyvers & Tenenbaum, 2005), and geometry-specific diffusion architectures have been proposed for graph generation (Wen et al., 2024; Fu et al., 2024), the role of latent curvature as an isolated inductive bias in a controlled generative setting has not been systematically examined. We instantiate this formulation using diffusion-based generative modeling defined intrinsically on the latent manifold. Node embeddings are sampled via Riemannian diffusion and converted into graphs through distance-based decoding. To support variable-size graphs, we integrate a hierarchical coarse-to-fine expansion strategy (Bergmeister et al., 2024). By isolating latent geometry from architectural complexity, the framework enables a systematic study of geometric inductive biases. We evaluate several latent spaces, including Euclidean, spherical, and hyperbolic ones, on synthetic graph families and the QM9 molecular dataset. Results show that generation quality depends strongly on alignment between latent geometry and graph structure, highlighting curvature as a principled mechanism for controlling inductive bias and generalization in graph generation.

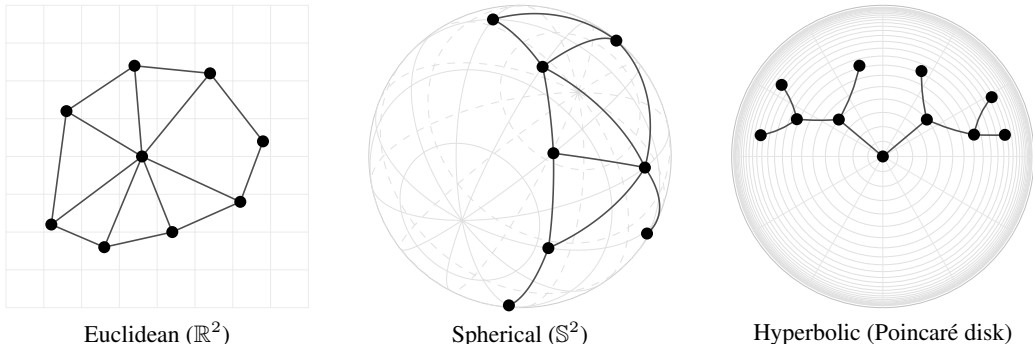

Euclidean ($\mathbb{R}^2$)  Spherical ($\mathbb{S}^2$)  Hyperbolic (Poincaré disk)

Figure 1: Inducing graphs from latent geometry. Nodes are embedded in a latent space whose metric defines structural inductive bias. Edges are obtained by thresholding distances $d(x_i, x_j) < \tau$, where $d$ is the metric induced by the chosen geometry.

## 2 Methodology

We formulate graph generation as a generative process over latent node embeddings defined on a geometric space. In this framework, discrete graph structure is not modeled directly but is induced from geometry: nodes are embedded in a latent manifold, and edges arise from geometric proximity. Generation is performed by sampling latent embeddings via a diffusion process and decoding graphs from the resulting geometric configuration. To support scalable generation of variable-size graphs, this process is coupled with a hierarchical expansion mechanism.

**Geometric Representation of Graphs.**  Let $G = (V, E)$ be an undirected, connected graph with $|V| = n$. Each node $i \in V$ is associated with a latent embedding $x_i \in \mathcal{M}$, where $(\mathcal{M}, g)$ is a Riemannian manifold equipped with metric $g$. The manifold geometry encodes inductive bias: curvature controls how distances expand and contract and thereby shapes which graph structures can be represented with low distortion. Graph structure is recovered from latent geometry through a distance-based decoding rule. Two nodes are connected if their geodesic distance is below a fixed threshold $\tau > 0$:

$$\{i, j\} \in E \quad \Longleftrightarrow \quad d_{\mathcal{M}}(x_i, x_j) < \tau,$$

where $d_{\mathcal{M}}$ denotes the geodesic distance induced by $g$. This decoding rule is intentionally minimal and is used consistently throughout training and generation, ensuring that all structural information must be expressed through geometry. The distance threshold $\tau$ is selected per dataset based on validation reconstruction accuracy and target edge density, and is kept fixed across all geometries to ensure a controlled comparison. We intentionally use a simple threshold-based decoder to isolate the effect of latent geometry rather than decoder expressivity.

**Embedding Generation.**  For each training graph, latent embeddings are obtained by directly optimizing node positions on $\mathcal{M}$ such that geodesic distances exactly reproduce adjacency relations. Given adjacency matrix $A \in \{0, 1\}^{n \times n}$, embeddings $X = \{x_1, \ldots, x_n\}$ are learned by minimizing

$$\mathcal{L}_{\text{embed}}(X) = \left\| A - \widehat{A}(X) \right\|_F^2, \qquad \widehat{A}_{ij}(X) = \mathbf{1}[d_{\mathcal{M}}(x_i, x_j) < \tau].$$

until the reconstruction matches the original graph. Optimization is performed using Riemannian gradient methods with projection to tangent spaces and retraction back onto the manifold. This procedure yields geometric configurations that serve as samples from the latent data distribution.

**Diffusion in Latent Geometry.**  To model the distribution of latent embeddings, we employ a score-based diffusion process defined intrinsically on the manifold $\mathcal{M}$, as illustrated in Figure 2. Because embeddings lie in curved spaces, noise injection, gradients, and updates must respect the underlying geometry and cannot be implemented by naively using standard Euclidean operations. We therefore define the forward noising process as Brownian motion on $\mathcal{M}$, which progressively removes geometric structure while remaining isotropic with respect to the manifold metric and converging to a simple reference distribution specific to the chosen geometry. The corresponding

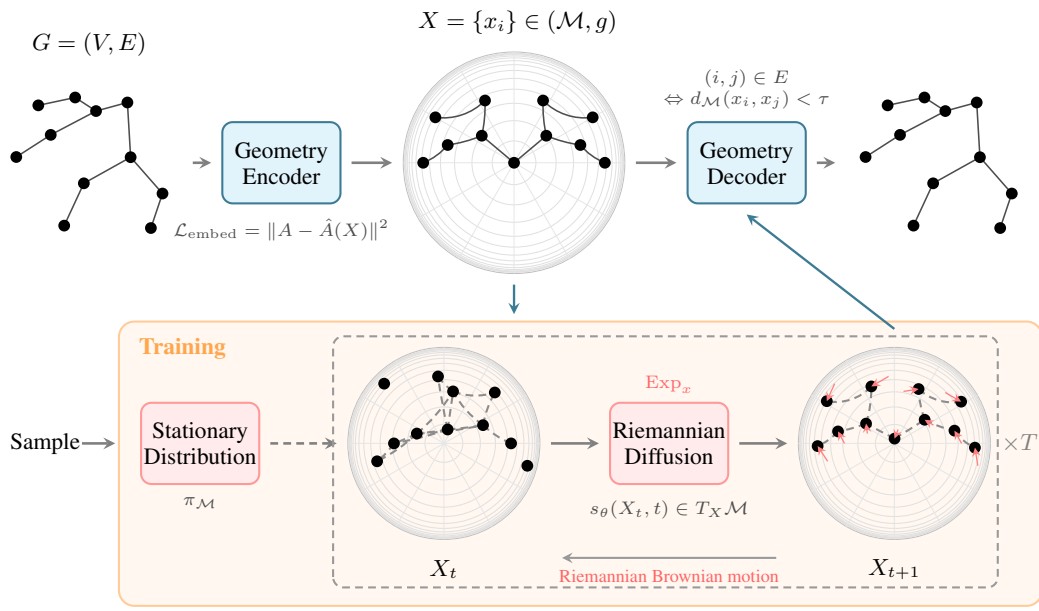

Figure 2: **(Top)** The encoder-decoder pipeline: a graph $G$ is embedded into a Riemannian manifold $(\mathcal{M}, g)$, and edges are recovered by thresholding pairwise distances. **(Bottom)** Training: samples from the stationary distribution $\pi_\mathcal{M}$ are iteratively denoised using a learned score function $s_\theta$, with updates performed via the Riemannian exponential map.

reverse-time dynamics define a generative process whose score is approximated by a neural network operating in tangent spaces. During training, noise is injected intrinsically in tangent spaces, denoising directions are predicted using a Riemannian score-matching objective, and updates are mapped back to the manifold via exponential maps, ensuring that all intermediate states remain valid points on $\mathcal{M}$. For non-Euclidean geometries, noise is injected in the tangent space of each embedding using the exponential and logarithmic maps, following standard manifold diffusion practice. Score matching is performed in the tangent space, and reverse-time sampling maps denoised samples back to the manifold. At inference time, generation is initialized by sampling latent embeddings from the stationary distribution of the forward diffusion process, corresponding to an unstructured geometric configuration. The learned intrinsic denoising operator is then applied iteratively, progressively transforming noise into a structured embedding arrangement through repeated local refinement, as depicted by the iterative loop in Figure 2. After the final denoising step, a graph is recovered by decoding edges directly from geometry using the distance-based thresholding rule described above. In this way, latent geometry, intrinsic diffusion dynamics, and geometric decoding jointly give rise to structured graph samples without explicit edge prediction.

**Hierarchical Graph Generation.** Directly applying diffusion to embeddings of large graphs requires the model to resolve both local and global structures at once. To enable scalable generation, we adopt a hierarchical coarse-to-fine expansion scheme based on graph coarsening, following the framework of Bergmeister et al. (2024). Each graph is associated with a hierarchy of progressively coarser graphs, obtained by contracting groups of nodes into supernodes. This process is invertible, allowing coarse representations to be deterministically expanded back to finer resolutions. During generation, the model operates at successive levels of this hierarchy. At each level, it predicts how coarse nodes expand into finer ones, while latent embeddings are replicated accordingly to preserve geometric consistency. Diffusion is applied locally at each level, and expansion proceeds iteratively until the target graph size is reached. Importantly, our method does not modify the hierarchical generation mechanism itself; instead, we focus on how different latent geometries interact with this expansion process. We therefore refer to Bergmeister et al. (2024) for full details of the hierarchical construction and decoding procedure.

Table 1: Synthetic graph generation results. We report validity (Valid), uniqueness (Unique), novelty (Novel), and their product (V.U.N.).

| Model | Planar | | | | SBM | | | | Tree | | | |
|---|---|---|---|---|---|---|---|---|---|---|---|---|
| | Valid | Unique | Novel | V.U.N. | Valid | Unique | Novel | V.U.N. | Valid | Unique | Novel | V.U.N. |
| GraphRNN (You et al., 2018) | 0.0 | 100 | 100 | 0.0 | 5.0 | 100 | 100 | 5.0 | — | — | — | — |
| GRAN (Liao et al., 2020) | 97.5 | 85.0 | 2.5 | 0.0 | 25.0 | 100 | 100 | 25.0 | 0.0 | 100 | 100 | 0.0 |
| SPECTRE (Martinkus et al., 2022) | 25.0 | 100 | 100 | 25.0 | 52.5 | 100 | 100 | 52.5 | — | — | — | — |
| DiGress (Vignac et al., 2023) | 77.5 | 100 | 100 | 77.5 | 60.0 | 100 | 100 | 60.0 | 90.0 | 100 | 100 | 90.0 |
| EDGE (Chen et al., 2023) | 0.0 | 100 | 100 | 0.0 | 0.0 | 100 | 100 | 0.0 | 0.0 | 7.5 | 100 | 0.0 |
| BwR (EDP-GNN) (Diamant et al., 2023) | 0.0 | 100 | 100 | 0.0 | 7.5 | 100 | 100 | 7.5 | 0.0 | 100 | 100 | 0.0 |
| BiGG (Dai et al., 2020) | 62.5 | 85.0 | 42.5 | 5.0 | 10.0 | 100 | 100 | 10.0 | 100 | 87.5 | 50.0 | 75.0 |
| GraphGen (Goyal et al., 2020) | 7.5 | 100 | 100 | 7.5 | 5.0 | 100 | 100 | 5.0 | 95.0 | 100 | 100 | 95.0 |
| LocalPGNN+ILE (Bergmeister et al., 2024) | 95.0 | 100 | 100 | 95.0 | 75.0 | 100 | 100 | 75.0 | 100 | 100 | 100 | **100** |
| Ours (Euclidean) | 97.5 | 100.0 | 100.0 | **97.5** | 77.5 | 100.0 | 100.0 | 77.5 | 90.8 | 100.0 | 100.0 | 90.8 |
| Ours ($\ell_\infty$) | 94.3 | 100.0 | 100.0 | 94.3 | 75.6 | 100.0 | 100.0 | 75.6 | 82.8 | 100.0 | 100.0 | 82.8 |
| Ours (Spherical) | 91.3 | 100.0 | 99.8 | 90.7 | 81.0 | 100.0 | 100.0 | **81.0** | 78.3 | 100.0 | 100.0 | 78.3 |
| Ours ($\ell_1$) | 85.2 | 99.2 | 100.0 | 84.9 | 71.2 | 100.0 | 100.0 | 71.2 | 81.5 | 100.0 | 100.0 | 81.5 |
| Ours (Hyperbolic) | 84.7 | 100.0 | 100.0 | 84.7 | 66.3 | 100.0 | 100.0 | 66.3 | 100.0 | 99.8 | 100.0 | **99.8** |

## 3  EXPERIMENTAL EVALUATION

We evaluate the impact of latent geometry on diffusion-based graph generation using three synthetic graph families from the SPECTRE benchmark (Martinkus et al., 2022): *Planar*, *Stochastic Block Model (SBM)*, and *Tree* graphs. These datasets exhibit distinct structural characteristics and serve as canonical test cases for Euclidean, positively curved, and negatively curved geometries, respectively. We further report results on the real-world QM9 molecular dataset; full details are provided in the Appendix. For synthetic graphs, we adopt the hierarchical local expansion framework of Bergmeister et al. (2024) as our primary baseline, as it matches our decoding paradigm and allows us to isolate the effect of latent geometry. Additional autoregressive and diffusion-based baselines follow the standard SPECTRE protocol. For QM9, we compare against DIGRESS (Vignac et al., 2023) and related molecule generators. We evaluate multiple latent geometries, including Euclidean, spherical, hyperbolic, $\ell_1$, and $\ell_\infty$ spaces. All other components, diffusion model, hierarchical expansion, and distance-based decoding, are kept fixed across geometries. Additional experimental results are provided in the appendix. In particular, we report an evaluation on the QM9 molecular dataset as well as an analysis of structural statistics for synthetic graphs, including degree distributions, clustering coefficients, and orbit counts. These results further support the main findings of the paper.

**Results for Synthetic Graphs.**  Table 1 reports validity, uniqueness, novelty, and their product (V.U.N.) under the standard SPECTRE evaluation protocol. Across all datasets, performance depends strongly on the alignment between latent geometry and graph structure. Euclidean embeddings perform best on planar graphs, spherical embeddings perform best on SBM graphs with multiple communities, and hyperbolic embeddings perform best on tree graphs. This consistent ordering confirms that curvature acts as an effective inductive bias for diffusion-based graph generation. More statistics and additional analyses are provided in the Appendix. The observed performance trends align with known geometric properties of graph families: tree-like graphs benefit from hyperbolic embeddings due to their exponential growth, planar graphs favor Euclidean geometry due to low distortion, and community-structured graphs benefit from bounded geometries that emphasize cluster separation.

## 4  CONCLUSION

We present a geometry-aware framework for diffusion-based graph generation in which graphs are generated by sampling latent node embeddings on metric spaces and decoding structure directly from geometry. By defining diffusion intrinsically on the latent manifold and combining it with hierarchical expansion, the approach separates structural inductive bias from architectural complexity. Our experiments demonstrate that the choice of latent geometry has a systematic and predictable impact on generative performance: Euclidean, spherical, and hyperbolic spaces perform best when aligned with the planar, community-based, and hierarchical structure of the target graph families, respectively. These results highlight latent geometry as a principled and effective mechanism for controlling structural bias in graph generative models, and suggest that geometric design choices can play a central role in graph generation.

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

# A   RELATED WORK

Diffusion models are a prominent class of generative models successfully applied to many data modalities, such as images, videos, and 3D geometric data (Yang et al., 2025). Graph generation itself is a well-studied task with applications in molecular design, floorplan synthesis, and program analysis (Shabani et al., 2022; Shi et al., 2020). Existing approaches can be broadly categorized into autoregressive models (Kong et al., 2023; You et al., 2018; Zhao et al., 2024; Jang et al., 2024) and one-shot generative models such as diffusion-based methods. Autoregressive methods offer flexibility in handling variable graph sizes and avoid the quadratic complexity characteristic of one-shot reconstruction, but they typically require a carefully chosen node ordering. In contrast, diffusion-based approaches generate entire graphs in a single denoising trajectory, removing the need for explicit orderings and enabling more global structural coherence.

Early diffusion models for graph generation relied on continuous-state diffusion processes, using score-based modeling to operate directly on node and edge feature matrices (Niu et al., 2020; Jo et al., 2022). Subsequent work expanded this line of research with diffusion mixtures (Jo et al., 2024) and latent continuous-state formulations such as Latent Graph Diffusion (LGD) (Zhou et al., 2024).Continuous Gaussian noise makes it straightforward to integrate heterogeneous graph attributes and has supported a range of applications in computational biology and molecular modeling, including diffusion-based docking methods (Corso et al., 2023). Hybrid methods further combine continuous diffusion for geometric coordinates with discrete diffusion over atom and bond types, enabling joint generation of molecular structures in 2D and 3D (Hua et al., 2024; Vignac et al., 2023).

Alongside continuous approaches, discrete-state diffusion models have become the dominant family of graph diffusion techniques. These models operate directly on categorical node and edge channels through discrete-time noising and denoising processes, as demonstrated in Haefeli et al. (2023). Among them, *DiGress* Vignac et al. (2023) stands out as a foundational framework: it provides a general-purpose, permutation-equivariant discrete diffusion mechanism that scales reliably, supports flexible factorization of node and edge channels, and serves as the backbone for numerous recent advances in both general-purpose and domain-specific graph generative modeling.

**Hyperbolic and Mixed-Curvature Diffusion Models for Graphs.**   Several recent works have explored non-Euclidean diffusion models specifically for graph generation. Wen et al. (2024) proposed HGDM, a two-stage model that first encodes graphs into hyperbolic node embeddings via a variational autoencoder and then trains score-based diffusion models that operate jointly in hyperbolic space for node representations and Euclidean space for adjacency structure, demonstrating improvements on graphs with pronounced hierarchical structure. Fu et al. (2024) introduced HypDiff, which defines anisotropic latent diffusion processes constrained by radial and angular geometric properties in hyperbolic space, preserving topological information during generation. Wang et al. (2024) proposed ProGDM, which extends graph diffusion to product spaces of mixed curvature via Riemannian embedding and geometric diffusion across multiple sub-spaces. These methods design new architectures or diffusion formulations tailored to specific geometries. In contrast, our work uses a fixed, deliberately simple pipeline and varies only the latent geometry, enabling a controlled comparison of how curvature alone influences generation quality across graph families.

**Hierarchical Methods for Graph Generation.**   A complementary line of work in graph generative modeling focuses on hierarchical or coarse-to-fine construction strategies. These approaches aim to improve scalability by decomposing the generation task into multiple levels of abstraction. Shirzad et al. (2022) propose a two-stage framework in which a tree-based cluster representation is first constructed via graph decomposition, followed by incremental subgraph refinement conditioned on this hierarchy. Davies et al. (2023) introduce HiGGs, a two-level method that first generates a coarse cluster graph using a discrete diffusion model (DiGress) before independently generating cluster subgraphs and intra-cluster edges. In a related direction, Karami (2023) develops HiGen, which extends hierarchical generation to multiple levels by autoregressively modeling cluster subgraphs. Limnios et al. (2023) propose SaGess, a divide-and-conquer strategy for accelerating discrete diffusion models such as DiGress by sampling coverings of subgraphs and denoising them in parallel. Hierarchical graph representations have also been studied outside of generative modeling in connection with GNNs, for instance in graph visualization (Grötschla et al., 2024).

Although these hierarchical schemes improve computational efficiency, they often rely on strong independence assumptions between clusters or hierarchical levels, which may limit fidelity for complex graph structure. Many such models also depend on externally computed clusterings; for example, both HiGGs and HiGen rely on the Louvain algorithm to construct cluster hierarchies during training. Beyond general graph generation, hierarchical techniques have also been widely developed in molecular domains, where structural motifs and domain knowledge naturally induce multi-scale organization. Examples include the Junction Tree VAE (Jin et al., 2018), spectral-preserving graph coarsening (Jin et al., 2020), and hierarchical normalizing flows for molecular generation such as MolGrow (Kuznetsov & Polykovskiy, 2021). While effective for chemically structured data, these molecular approaches are not directly optimized for general-purpose graph generation.

**Non-Euclidean Geometry for Graphs.**    A substantial body of empirical evidence indicates that many real-world graphs exhibit structural properties that are fundamentally non-Euclidean. Networks arising in domains such as social systems, biology, and knowledge representation commonly display hierarchical organization, heterogeneous community structure, heavy-tailed degree distributions, and tree-like branching patterns. Representing these characteristics in a flat Euclidean space often leads to significant distortion unless high-dimensional embeddings are used, and even then the Euclidean metric may fail to reflect meaningful relational structure.

Manifold-based approaches provide a principled alternative by embedding nodes in spaces whose curvature better matches the topology of the underlying graph. Prior work across scientific fields emphasizes that various data modalities naturally reside on curved manifolds rather than in $\mathbb{R}^n$, including graph-structured and hierarchical data (Roy et al., 2006; Steyvers & Tenenbaum, 2005), climate and geoscience data on the sphere (Karpatne et al., 2019; Mathieu & Nickel, 2020), and structural or geometric data in SE(3) (Jumper et al., 2021; Simeonov et al., 2022). Although these examples span diverse domains, they collectively underscore a broader point: when data possess intrinsic non-Euclidean structure, Euclidean representations can be ill-suited, potentially distorting the geometry that encodes salient relationships.

**Hyperbolic Embeddings and Networks.**    The utility of hyperbolic geometry for representing hierarchical and tree-like data has been established through a line of foundational work. Ganea et al. (2018) introduced hyperbolic versions of core neural network operations, including feed-forward and recurrent layers, in the Poincaré ball model. Chami et al. (2019) proposed HGCN, the first inductive hyperbolic graph convolutional network, which maps Euclidean input features to hyperbolic embeddings with trainable curvature at each layer and demonstrated significant improvements on link prediction and node classification for hierarchical graphs. These works, among others, established the practical foundations for non-Euclidean deep learning on graphs and motivate the use of hyperbolic latent spaces in our generative framework.

**Latent Geometry Selection for Graphs.**    A growing body of work studies how the choice of latent geometry affects graph representation and inference. Kazi et al. (2022) introduced the Differentiable Graph Module (DGM), which learns a latent graph from data in an end-to-end fashion using edge probabilities derived from embeddings in a learned latent space. de Ocáriz Borde et al. (2023) extended this framework to product manifolds of constant curvature model spaces, showing that richer latent geometries can improve the quality of inferred graphs. Sáez de Ocáriz Borde et al. (2023) formalized the problem of neural latent geometry search (NLGS), proposing a Bayesian optimization procedure over product manifold signatures informed by the Gromov-Hausdorff distance. More recently, de Ocáriz Borde & Kratsios (2023) introduced neural snowflakes, a trainable architecture that implements fractal-like metrics and provably admits isometric embeddings of arbitrary finite weighted graphs, removing the need for combinatorial search over geometry families. In a related direction, Sun et al. (2022) proposed a self-supervised mixed-curvature GNN that constructs product spaces via Cartesian products of Riemannian components and uses contrastive learning across curvature views. These works focus primarily on discriminative tasks such as node classification and link prediction. Our work complements this line of research by studying how fixed latent geometries of different curvatures influence *generative* modeling of graph distributions, rather than optimizing geometry for a downstream prediction task.

# B ADDITIONAL EXPERIMENTAL RESULTS

## B.1 STRUCTURAL STATISTICS OF GENERATED GRAPHS

To complement validity-based evaluation, we compare distributions of graph statistics between generated graphs and the training data, following the GraphRNN protocol (You et al., 2018). We report distances between degree distributions (Deg.), clustering coefficient distributions (Clus.), and 4-node orbit motif distributions (Orbit). Table 2 summarizes these results.

For graph families with strong local constraints, these statistics track validity closely. On planar graphs, orbit statistics are particularly informative because non-planarity typically manifests through dense local substructures; Euclidean embeddings match orbit distributions most closely and achieve the highest validity. On tree graphs, clustering and orbit metrics are highly constrained, and hyperbolic embeddings match the corresponding statistics nearly exactly. In contrast, on SBM graphs, local statistics are less predictive of validity because the defining structure is global (community separation); accordingly, differences between geometries are more visible in validity than in local distributional metrics.

## B.2 MOLECULAR GRAPH GENERATION ON QM9

Table 3 reports validity, uniqueness, novelty, and V.U.N. on QM9. Unlike most molecular generators, our model does not use atom types, bond types, or chemical features: molecules are treated as unlabeled graphs, and validity reflects only topological correctness (e.g., connectedness and degree constraints) rather than chemical plausibility. Stability metrics and property-based evaluations, which are standard in the molecular generation literature, are not applicable in this feature-agnostic setting. The QM9 results therefore serve to evaluate the effect of latent geometry on graph-level structural validity rather than as a comparison with full-featured molecular generators. Within this feature-agnostic setting, Euclidean and hyperbolic geometries achieve high validity, while spherical geometry performs substantially worse.

These results suggest that, for QM9, locally flat or mildly negatively curved latent spaces provide stable geometric representations for the local bonding patterns present in the dataset. Conversely, strong positive curvature can impose global constraints that conflict with common molecular motifs, reducing validity.

Table 2: Graph Generation Statistics

| Planar Graphs ($n_{max}$ = 64, $n_{avg}$ = 64) | | | | |
|---|---|---|---|---|
| **Model** | **Deg.↓** | **Clus.↓** | **Orbit↓** | **Valid↑** |
| Training set | 0.0002 | 0.0310 | 0.0005 | |
| GraphRNN (You et al., 2018) | 0.0049 | 0.2779 | 1.2543 | 0.0 |
| GRAN (Liao et al., 2020) | 0.0007 | 0.0426 | 0.0009 | 97.5 |
| SPECTRE (Martinkus et al., 2022) | 0.0005 | 0.0785 | 0.0012 | 25.0 |
| DiGress (Vignac et al., 2023) | 0.0007 | 0.0780 | 0.0079 | 77.5 |
| EDGE (Chen et al., 2023) | 0.0761 | 0.3229 | 0.7737 | 0.0 |
| BwR (EDP-GNN) (Diamant et al., 2023) | 0.0231 | 0.2596 | 0.5473 | 0.0 |
| BiGG (Dai et al., 2020) | 0.0007 | 0.0570 | 0.0367 | 62.5 |
| GraphGen (Goyal et al., 2020) | 0.0328 | 0.2106 | 0.4236 | 7.5 |
| LocalPGNN+ILE (Bergmeister et al., 2024) | 0.0005 | 0.0626 | 0.0017 | 95.0 |
| Ours (Euclidean) | 0.0006 | 0.0505 | 0.0009 | 97.5 |
| Ours ($L_\infty$) | 0.0006 | 0.0593 | 0.0020 | 94.3 |
| Ours (Spherical) | 0.0006 | 0.0549 | 0.0025 | 91.3 |
| Ours (L1) | 0.0007 | 0.0512 | 0.0054 | 85.2 |
| Ours (Hyperbolic) | 0.0007 | 0.0680 | 0.0052 | 84.7 |

| Stochastic Block Model ($n_{max}$ = 187, $n_{avg}$ = 104) | | | | |
|---|---|---|---|---|
| **Model** | **Deg.↓** | **Clus.↓** | **Orbit↓** | **Valid↑** |
| Training set | 0.0008 | 0.0332 | 0.0255 | — |
| GraphRNN (You et al., 2018) | 0.0055 | 0.0584 | 0.0785 | 5.0 |
| GRAN (Liao et al., 2020) | 0.0113 | 0.0553 | 0.0540 | 25.0 |
| SPECTRE (Martinkus et al., 2022) | 0.0015 | 0.0521 | 0.0412 | 52.5 |
| DiGress (Vignac et al., 2023) | 0.0018 | 0.0485 | 0.0415 | 60.0 |
| EDGE (Chen et al., 2023) | 0.0279 | 0.1113 | 0.0854 | 0.0 |
| BwR (EDP-GNN) (Diamant et al., 2023) | 0.0478 | 0.0638 | 0.1139 | 7.5 |
| BiGG (Dai et al., 2020) | 0.0012 | 0.0604 | 0.0667 | 10.0 |
| GraphGen (Goyal et al., 2020) | 0.0550 | 0.0623 | 0.1189 | 5.0 |
| LocalPGNN+ILE (Bergmeister et al., 2024) | 0.0141 | 0.0528 | 0.0809 | 75.0 |
| Ours (Spherical) | 0.0190 | 0.0548 | 0.0968 | 81.0 |
| Ours (Euclidean) | 0.0162 | 0.0536 | 0.0874 | 77.5 |
| Ours ($L_\infty$) | 0.0146 | 0.0531 | 0.0825 | 75.6 |
| Ours (L1) | 0.0110 | 0.0517 | 0.0711 | 71.2 |
| Ours (Hyperbolic) | 0.0070 | 0.0502 | 0.0581 | 66.3 |

| Tree Graphs ($n_{max}$ = 64, $n_{avg}$ = 64) | | | | |
|---|---|---|---|---|
| **Model** | **Deg.↓** | **Clus.↓** | **Orbit↓** | **Valid↑** |
| Training set | 0.0001 | 0.0000 | 0.0000 | — |
| GRAN (Liao et al., 2020) | 0.1884 | 0.0080 | 0.0199 | 0.0 |
| DiGress (Vignac et al., 2023) | 0.0002 | 0.0000 | 0.0000 | 90.0 |
| EDGE (Chen et al., 2023) | 0.2678 | 0.0000 | 0.7357 | 0.0 |
| BwR (EDP-GNN) (Diamant et al., 2023) | 0.0016 | 0.1239 | 0.0003 | 0.0 |
| BiGG (Dai et al., 2020) | 0.0014 | 0.0000 | 0.0000 | 75.0 |
| GraphGen (Goyal et al., 2020) | 0.0105 | 0.0000 | 0.0000 | 95.0 |
| LocalPGNN+ILE (Bergmeister et al., 2024) | 0.0001 | 0.0000 | 0.0000 | **100** |
| Ours (Hyperbolic) | 0.0000 | 0.0000 | 0.0000 | 100.0 |
| Ours (Euclidean) | 0.0032 | 0.0000 | 0.0000 | 90.8 |
| Ours ($L_\infty$) | 0.0030 | 0.0000 | 0.0000 | 82.8 |
| Ours (L1) | 0.0039 | 0.0000 | 0.0000 | 81.5 |
| Ours (Spherical) | 0.0012 | 0.0000 | 0.0000 | 78.3 |

Table 3: QM9 Dataset

| Model | Valid↑ | Unique↑ | Novel↑ | V.U.N↑ |
|---|---|---|---|---|
| Set2GraphVAE (Serviansky et al., 2020) | 59.9 | 93.8 | — | — |
| SPECTRE (Martinkus et al., 2022) | 87.3 | 35.7 | — | — |
| GraphNVP (Madhawa et al., 2019) | 83.1 | 99.2 | — | — |
| GDSS (Jo et al., 2022) | 95.7 | 98.5 | — | — |
| ConGress (Vignac et al., 2023) | 98.9 | 96.8 | — | — |
| DiGress (Vignac et al., 2023) | 99.0 | 96.2 | — | — |
| Ours (Euclidean) | $97.8 \pm 1.3$ | $100.0 \pm 0.0$ | $100.0 \pm 0.0$ | $97.8 \pm 1.3$ |
| Ours (Hyperbolic) | $97.2 \pm 0.6$ | $100.0 \pm 0.0$ | $100.0 \pm 0.0$ | $97.2 \pm 0.6$ |
| Ours ($L_\infty$) | $94.8 \pm 2.8$ | $100.0 \pm 0.0$ | $100.0 \pm 0.0$ | $94.8 \pm 2.8$ |
| Ours (L1) | $92.3 \pm 1.0$ | $100.0 \pm 0.0$ | $100.0 \pm 0.0$ | $92.3 \pm 1.0$ |
| Ours (Spherical) | $87.0 \pm 1.0$ | $100.0 \pm 0.0$ | $100.0 \pm 0.0$ | $87.0 \pm 1.0$ |

