# OpenReview forum: "Geometric Inductive Biases for Diffusion-Based Graph Generation"
_ICLR.cc/2026/Workshop/GRaM — ICLR 2026 Workshop GRaM Poster_

### Official Review · Reviewer_37Kt · 2026-02-20

**Rating:** 6
**Confidence:** 5

**Review:**

This paper studies diffusion-based graph generation from latent geometries of different curvature and shows that performance improves when the latent geometry matches graph structure. The framework is conceptually clean and fits the workshop theme, but it is largely an empirical study built from standard components rather than a new generative model.
Several concerns remain. First, the QM9 evaluation treats molecules as unlabeled graphs and does not report stability metrics, making it unclear how meaningful the results are for molecular generation. Second, the approach relies on pre-optimizing graph embeddings in each geometry, which may encode strong supervision and reduce the difficulty of the generative task. Third, the decoder is a simple distance threshold, so improvements may largely reflect embedding reconstruction quality rather than generative modeling capacity. Finally, computational cost and scalability of manifold diffusion and embedding optimization are not discussed.
Overall, the paper provides an interesting perspective on geometric inductive bias but is primarily exploratory and limited in technical depth.

**Pmlr Suitability:**

NA

---

### Official Review · Reviewer_7xav · 2026-02-21

**Rating:** 4
**Confidence:** 5

**Review:**

The paper frames graph generation as sampling node embeddings in a latent metric space and decoding edges via a distance threshold.

The novelty claim around manifold latent geometries as an inductive bias for graph generation is not clearly supported by the current draft. A substantial body of prior work on latent graph inference, transformers, knowledge graphs, and geometry selection (including product-manifold and learnable latent geometries) shares a similar motivation and explores overlapping geometric families, often generalizing beyond the specific choices considered here. See:

- Hyperbolic Graph Generator

- Differentiable Graph Module (DGM) for Graph Convolutional Networks

- Latent Graph Inference using Product Manifolds

- Curve Your Attention: Mixed-Curvature Transformers for Graph Representation Learning

- Neural Latent Geometry Search: Product Manifold Inference via Gromov-Hausdorff-Informed Bayesian Optimization

- A Self-supervised Mixed-curvature Graph Neural Network

- Mixed-Curvature Multi-Relational Graph Neural Network for Knowledge Graph Completion

- Mixed-curvature knowledge-enhanced graph contrastive learning for recommendation

- Neural Snowflakes: Universal Latent Graph Inference via Trainable Latent Geometries

- Neural Spacetimes for DAG Representation Learning

These works also suggest several directions that could naturally carry over to diffusion settings.

The paper would benefit from explicitly positioning against these works and clarifying what is new relative to them. As written, the key contribution appears to be intrinsic manifold diffusion for latent embeddings within a graph-generation pipeline, rather than the geometry choice itself.

Moreover, diffusion-based graph generation has substantial prior work (which the related-work section partially reflects). There is also relevant prior work on hyperbolic and mixed-curvature manifold diffusion models for graphs that is not discussed, e.g.:

- Hyperbolic Graph Diffusion Model

- Hyperbolic Geometric Latent Diffusion Model for Graph Generation

- A Mixed-Curvature Graph Diffusion Model

Additionally, the paper would also benefit from mentioning early works in hyperbolic embeddings and networks:

- Hyperbolic Neural Networks
- Hyperbolic Neural Networks++
- Hyperbolic Entailment Cones for Learning Hierarchical Embeddings
- Poincaré GloVe: Hyperbolic Word Embeddings
- Hyperbolic Graph Neural Networks: A Review of Methods and Applications

Overall, without clearer positioning and differentiation, the contribution currently reads as incremental relative to the cited prior work. If the authors intend the main novelty to be the geometry component, an ablation or comparison that incorporates stronger geometry-selection baselines (e.g., product manifolds or trainable geometry families such as snowflakes) would help substantiate that claim.

**Pmlr Suitability:**

NA

---

### Meta-Review · Area_Chair_8Brz · 2026-02-25

**Decision:**

Accept

**Metareview:**

It explores geometric inductive biases for diffusion on graphs using mixed-curvature manifolds. Even though it's built on somewhat standard components and misses some related work, it's a solid and correct empirical exploration that fits the tiny paper track well. I thus conclude to accept.

**Relevance To Proceedings:**

Tiny paper — does not apply

**Relevance To Workshop:**

Yes — suitable for GRaM

---

### Decision · Program_Chairs · 2026-03-02

Accept (Poster)